# Performance of the Fecal Immunochemical Test in Detecting Advanced Colorectal Neoplasms and Colorectal Cancers in People Aged 40–49 Years: A Systematic Review and Meta-Analysis

**DOI:** 10.3390/cancers15113006

**Published:** 2023-05-31

**Authors:** Jen-Hao Yeh, Cheng-Hao Tseng, Wen-Lun Wang, Chih-I Chen, Yu-Peng Liu, Yi-Chia Lee, Jaw-Yuan Wang, Yu-Ching Lin

**Affiliations:** 1Division of Gastroenterology and Hepatology, Department of Internal Medicine, E-DA DaChang Hospital, I-Shou University, Kaohsiung 813, Taiwan; b9202078@gmail.com; 2Department of Medical Technology, College of Medicine, I-Shou University, Kaohsiung 824, Taiwan; 3Graduate Institute of Clinical Medicine, Kaohsiung Medical University, Kaohsiung 813, Taiwan; ypliu@kmu.edu.tw; 4Division of Gastroenterology and Hepatology, Department of Internal Medicine, E-DA Cancer Hospital, Kaohsiung 824, Taiwan; chenhaug@gmail.com (C.-H.T.); warrengodr@gmail.com (W.-L.W.); 5Division of Colorectal Surgery, Department of Surgery, E-DA Hospital, Kaohsiung 824, Taiwan; jimmyee0901@gmail.com; 6Research Center for Environmental Medicine, Kaohsiung Medical University, Kaohsiung 813, Taiwan; 7Division of Gastroenterology and Hepatology, Department of Internal Medicine, National Taiwan University Hospital, Taipei 100, Taiwan; yichialee@ntu.edu.tw; 8Division of Colorectal Surgery, Department of Surgery, Kaohsiung Medical University Hospital, Kaohsiung Medical University, Kaohsiung 813, Taiwan; 9Department of Surgery, Faculty of Medicine, College of Medicine, Kaohsiung Medical University Hospital, Kaohsiung Medical University, Kaohsiung 813, Taiwan; 10Center for Cancer Research, Kaohsiung Medical University, Kaohsiung 813, Taiwan; 11Pingtung Hospital, Ministry of Health and Welfare, Pingtung 900, Taiwan; 12Department of Family Medicine, E-DA DaChang Hospital, I-Shou University, Kaohsiung 824, Taiwan

**Keywords:** fecal immunochemical test, aged 40–49 years, colorectal cancer, advanced colorectal neoplasm

## Abstract

**Simple Summary:**

Recently, the cases of colorectal cancers has been rising in younger age (<50) individuals. Although current guidelines recommend colorectal cancer screening should be initiated at age 45 instead of 50, the optimal approach of colorectal cancer screening is not clear. This article investigates the efficacy of fecal immunochemical test (FIT), which detects occult blood in stool, in predicting advanced colorectal polyps and tumors among people aged 40–49. The findings suggest FIT is useful to identify such people with high risk to have advanced colorectal lesions. Hence, FIT may be considered as the first-line screening tool for these people, and further comparative study between FIT and colonoscopy will be of great value.

**Abstract:**

Background: The incidence of early-onset colorectal cancer (CRC) is increasing. Many guidelines recommend initiating screening at 45 years. This study investigated the detection rate of advanced colorectal neoplasm (ACRN) by using fecal immunochemical tests (FITs) in individuals aged 40–49 years. Methods: PubMed, Embase, and Cochrane Library databases were searched from inception to May 2022. The primary outcomes were the detection rates and positive predictive values of FITs for ACRN and CRC in people aged 40–49 (younger age group) and ≥50 years (average risk group). Results: Ten studies with 664,159 FITs were included. The FIT positivity rate was 4.9% and 7.3% for the younger age and average risk groups, respectively. Younger individuals with positive FIT results had significantly higher risks of ACRN (odds ratio [OR] 2.58, 95% confidence interval [CI] 1.79–3.73) or CRC (OR 2.86, 95% CI 1.59–5.13) than did individuals in the average-risk group, regardless of FIT results. Individuals aged 45–49 years with positive FIT results had a similar risk of ACRN (OR 0.80, 95% CI 0.49–1.29) to that of people aged 50–59 years with positive FIT results, although significant heterogeneity was observed. The positive predictive values of the FIT were 10–28.1% for ACRN and 2.7–6.8% for CRC in the younger age group. Conclusion: The detection rate of ACRN and CRC based on FITs in individuals aged 40–49 years is acceptable, and the yield of ACRN might be similar between individuals aged 45–49 and 50–59 years. Further prospective cohort and cost-effective analysis are warranted.

## 1. Introduction

Colorectal cancer (CRC) is a leading cause of mortality worldwide [1,2,3]. Because of organized screening programs, the overall incidence and mortality of CRC have begun to decline in developed countries [4,5,6]. However, a persistent trend of new CRC cases among individuals under 50 years of age has been observed [7,8]. Incidence has nearly doubled in both Western and Asian countries in two decades and has especially increased in individuals aged younger than 40 years [9,10].

In addition to this rapidly rising incidence, early-onset colorectal cancer (EOCRC), CRC occurring before 50 years of age, has become a particular public health problem with several characteristics. Compared with normal CRC, EOCRC tends to be located on the left side of the colon and the rectum [11], be present in a higher proportion of individuals with a first-degree family history or hereditary cancer syndrome [12,13], and be identified at an advanced stage because of delayed diagnosis [14]. This implies a necessity for prior screening recommendations focused on individuals aged ≥50 years.

Several society guidelines recommend an earlier screening age of 45 years in response to increasing EOCRC [15,16,17,18]. However, these recommendations are mostly based on colonoscopy-based screening, which has high costs and personnel burden. Fecal immunochemical tests (FITs) are another widely accepted CRC screening modality with lower costs and invasiveness than colonoscopy. In people aged ≥50 years, the efficacy of annual to biannual FIT is similar to that of colonoscopy [19,20]. Because of the lower absolute incidence of CRC in younger people than in those aged ≥50 years [6], FITs are theoretically a promising alternative screening tool for EOCRC. However, the efficacy of FITs in younger individuals is unclear [16]. Therefore, a systematic review of available studies would be valuable. This study investigated the efficacy of FITs for younger individuals in the detection of CRC and advanced colorectal neoplasm (ACRN).

## 2. Methods

### 2.1. Search Strategy

The literature searches were conducted on PubMed, Embase, and Cochrane Library from inception to May 2022. The keywords and search strategy are described in Appendix A. The study was prospectively registered to the PROSPERO database (No. CRD42022333124). Two authors (JHY and YCL) independently and manually reviewed and identified eligible studies per prespecified criteria, and a consensus was reached through discussion. When disagreements were unresolved, the corresponding author JYW made the final decision.

### 2.2. Selection Criteria and Data Extraction

Studies were eligible if they investigated colorectal polyps or cancers in individuals aged 40–49 years who received FITs. Studies were excluded if (1) they lacked colonoscopy outcome data, (2) were simulation model studies, (3) had overlapping cohorts with other studies, or (4) were non-English language studies. The following data were then extracted from eligible studies: the name of the first author, year of publication, size, and characteristics of the studies population, FIT brand name, and colonoscopy results. All data were extracted as originally stated or after appropriate calculations. If the necessary data were unavailable, we contacted the corresponding author for additional information.

### 2.3. Outcome Assessment

The primary outcomes were the detection rate and positive predictive value (PPV) of FITs for ACRN and CRC. ACRN consisted of CRC with adenomatous polyps with high-risk features such as a size of ≥1 cm, high-grade dysplasia, or a villous component. The secondary outcomes were the detection rate and PPV for overall colorectal neoplasm (CRN), which included low-risk adenomas and ACRN. Comparisons were made between the younger age (40–49 years) and average-risk groups (≥50 years).

### 2.4. Statistical Analysis

Comprehensive Meta-Analysis version 3.3.070 (Biostat, Englewood, NJ, USA) with random-effects models were used for all meta-analyses. Odds ratios (ORs) were used for the analysis of the categorical outcome variables. Corresponding 95% confidence intervals (CIs) were used to compare the outcomes of the younger age and average-risk groups. The pooled effect size was significant if the ranges of the 95% CIs of the ORs excluded 1. The I^2^ statistic was used to evaluate statistical heterogeneity, which was considered significant if I^2^ > 50% or if a chi-square test yielded a *p*-value of <0.1.

### 2.5. Sensitivity Analysis and Risk of Bias Assessment

For all meta-analyses, we evaluated the robustness of the pooled effect estimates by excluding one study at a time. The risk of bias was assessed using the Newcastle–Ottawa scale for retrospective studies. We did not calculate publication bias in this study because the analysis was likely to be underpowered due to relatively few studies in the meta-analyses. The review adhered to Preferred Reporting Items for Systematic Reviews and Meta-Analyses (PRISMA) guidelines [21] (Appendix A).

## 3. Results

### 3.1. Baseline Characteristics of the Included Studies and Patients

The search strategy generated 4085 records and 186 potentially eligible studies. After the manual review process, 10 studies were included in the final analysis (Figure 1). The risk of bias assessment indicated that all studies included in this review were of high quality. (Appendix A). All enrolled studies were retrospective cohort analyses, most of which included individuals who participated in health checkups or CRC screening programs, except for that of D’Souza et al. [22], which focused on patients with symptoms who took FITs. Colonoscopy was offered to all participants in some studies [23,24,25,26,27] but only for people with positive FIT results in others [22,28,29,30,31]. Adenoma detection for the average-risk group ranged from 28.2% to 33% [22,24,26,27].

In total, 664,159 FITs (35.6% were younger-age group) were included in the studies, and the pooled FIT positivity rate was 4.9% for the younger age group and 7.3% for the average-risk group. Most of the studies used FITs from the OC-SENSOR series [24,25,26,27,28,29,30,31] (Eiken Chemical, Tokyo, Japan); Magstream/Hem SP [23] (Fujirebio, Tokyo, Japan) and HM-Jack analyzer [22] (Hitachi Chemical Diagnostics Systems, Tokyo, Japan) models were the next most common. The cutoff FIT values were either 10 or 20 µg Hb/g feces (Table 1).

### 3.2. Risk of Colorectal Neoplasia by Age and FIT Results

The characteristics of colorectal neoplasms by FIT result are presented in Table 2. A total of seven studies [22,24,25,26,27,28,30] reported ACRN detection rates and six studies [22,24,25,27,28,30] reported CRC detection rates stratified by FIT result and age, respectively. Pooled analysis revealed that ACRN was significantly more common among individuals with positive FIT results than in those with negative FIT results in both the younger-age (OR 9.98, 95% CI 5.98–16.67) and average-risk groups (OR 10.61, 95% CI 7.17–15.70). Although the risk of ACRN was higher in younger individuals with positive FIT results than among all average-risk individuals (OR 2.58, 95% CI 1.79–3.73, Figure 2), it was lower than that of average-risk individuals with positive FIT results (OR 0.54, 95% CI 0.34–0.86, Appendix A). Sensitivity analysis suggested the ACRN detection rate would have been similar in younger-age and average-risk individuals with positive FIT results if the study with high patient numbers (Chen et al.) [25] had been excluded (OR 0.57, 95% CI 0.32–1.02, Appendix A).

Likewise, CRC was significantly more common among individuals with positive FIT results than among those with negative FIT results in both the younger age (OR 19.6, 95% CI 10.17–37.78) and average-risk groups (OR 36.41, 95% CI 15.24–86.98). Younger individuals with positive FIT results had a higher risk of CRC than did all average-risk individuals (OR 2.86, 95% CI 1.59–5.13, Figure 3). A comparison between younger and average-risk individuals with positive FIT results suggested that younger individuals had a lower risk of CRC (OR 0.38, 95% CI 0.28–0.53, Appendix A) and lower risk of overall CRN (OR 0.50, 95% CI 0.36–0.70, Appendix A). However, overall CRN was significantly more common among younger individuals with positive FIT results than among average-risk individuals (OR 2.46, 95% CI 1.57–3.87).

### 3.3. Colorectal Neoplasia Risk by FIT in Individuals Aged 45–49 and 50–59 Years

Because the literature suggests similar screening yields for individuals aged 45–49 years and their older counterparts [7,32,33], we investigated the efficacy of FITs in individuals aged 45–49 and 50–59 years on the basis of ACRN and CRC risk. Among the four studies available for analysis, one included CRC data only [25], and three reported the overall risk of ACRN (Jung et al. did not report CRC case details) [26,27,30]. The pooled analysis demonstrated that the risk of ACRN was similar between individuals with positive FIT results aged 45–49 and 50–59 years (OR 0.80, 95% CI 0.49–1.29, Figure 4). However, sensitivity analysis suggested the ACRN detection rate would have been lower for individuals aged 45–49 years if the study by Levin et al [30]. had been excluded (OR 0.62, 95% CI 0.48–0.80, Appendix A). In individuals with positive FIT results, those aged 45–49 years had a lower risk of CRC than those aged 50–59 years (OR 0.59, 95% CI 0.45–0.79, Figure 5).

### 3.4. Performance of FIT by Age, Lesion Type, and Cutoff Value

The PPV of FITs was substantially affected by the severity of lesions (small/advanced adenomas and CRC) and varied by age and cutoff value (Appendix A). For ACRN, the PPV of FITs ranged from 10–28.1% for younger individuals and 21.3–30.9% for average-risk individuals. Negative predictive values were generally higher than 96% for younger individuals [22,24,26,27].

In the three studies that reported CRC cases (Appendix A) [22,27,31], FITs had a 2.7–6.8% PPV and a ≥99.5% negative predictive value for younger individuals. Both Yeh et al. [27] and Jung et al. [26] demonstrated inaccurate results of FITs for individuals younger than 30 years, in whom positive FIT results were not associated with higher ACRN detection. D’Souza et al. suggested that a lower FIT threshold could lead to substantially greater screening required for CRC detection [22]. With the cutoff adjusted from 150 μg/g to a detection limit of 2 μg/g of feces, the number of screenings necessary for the detection of one CRC case increased from 8.8 to 23.8 for individuals aged ≥ 50 years and increased from 2.9 to 10.9 for individuals aged < 50 years. Although the PPV increased with a higher threshold, the negative predictive value for CRC was 99% in both age groups for FIT thresholds of 2, 10, or 150 μg/g of feces.

## 4. Discussion

CRC screening programs using either colonoscopy or fecal occult blood tests are useful in lowering incidence and mortality [2,5,34,35]. FITs are the fecal occult blood test of choice because of their superior diagnostic accuracy, convenience, and cost-effectiveness over stool guaiac tests [36,37,38]. Although lowering the age for screening in response to the increasing threat of EOCRC is reasonable, the optimal strategy for younger individuals is not fully understood. Compared with individuals aged 50 years or older, younger individuals have a lower absolute prevalence of ACRN and CRC [1,6,7,14], which might make FITs an attractive option because of their lower cost and higher accessibility. Therefore, a rigorous evaluation of the diagnostic efficacy of FITs would be beneficial to determining the optimal EOCRC screening and prevention approach.

In this study, positive FIT results indicated a significantly higher risk of ACRN and CRC for people aged 40–49 years. Although the PPV of FITs in this population was lower than that of older individuals, the negative predictive value for CRC was sufficiently high in both age groups to be considered a first-line screening tool. In most studies with healthy participants, sensitivity to ACRN is less than 30% and is even lower for smaller colorectal adenomas. However, this lower sensitivity can be overcome with periodic checkups, similar to screening for average-risk individuals. FITs have lower financial and staff requirements than colonoscopies. Because of the lower absolute incidence of CRC in younger individuals, FITs may be a reasonable modality, with a balance between efficacy and cost.

Considering the appropriate timing and possible high-risk candidates for FIT screening for EOCRC is essential. Although guidelines recommend a screening age of 45 years, the subgroup analysis in our study revealed that individuals aged 45–49 years with positive FIT results have a lower risk of CRC than individuals aged 50–59 years with positive FIT results. However, the pooled analysis demonstrated that ACRN incidence might be similar in both age groups with positive FIT results, despite significant statistical heterogeneity. This result might be explained by differences in ethnicity in the younger age group. In this analysis, the outlier study (Levin et al.) [30] only included African American individuals in the 45–49 year-old group, and the other three studies included Asian individuals. African American individuals may have higher CRC incidence and mortality because of lower accessibility to screening programs [39,40]. Being a noninvasive test, the FIT has the potential as a complementary component with colonoscopy in screening programs.

Additional cost-effectiveness and risk stratification studies are warranted. Our study suggests that a universal FIT-based screening strategy for individuals aged 40–49 years may have a lower yield than the screening of average-risk individuals. To improve diagnostic yielding, the incorporation of other risk factors before enrollment in screening may be helpful. Multiple EOCRC risk factors such as Westernized diets, stress, antibiotic use, a sedentary lifestyle, and first-degree family history [41] have been identified, but few have been incorporated into screening considerations. Yeh et al. [27] suggested that metabolic syndrome and fatty liver disease are key risk factors for ACRN in younger individuals who receive FITs. Moreover, Levin et al. [30] is the only included study focused on African American individuals, and they found the screening yield was similar among ages 45–50 years African Americans and age 51–56 whites, Hispanics, and Asian-Pacific islanders. This finding implies race may be an important consideration for EOCRC screening. These factors, and other modifiable factors such as tobacco use and obesity, should be further evaluated in subsequent studies and considered as targets for screening and intervention.

This study has several strengths. First, it is a systematic review with integrated analysis from up-to-date cohort studies of FIT screening for younger individuals. Moreover, the meta-analyses provides an accurate estimation of the diagnostic efficacy of FIT in these population. Second, the differences related to the age and characteristics of the participants were analyzed. However, this review has several limitations. First, the earliest age for FIT screening in most studies was 40 years; thus, the results cannot be extrapolated to individuals aged younger than 40 years. Among three studies included individuals aged younger than 40 years, the median age of the younger group was 44 years in D’Souza et al. (NICE FIT study) [22], whereas Symonds et al. and Pin-Vieito et al. did not mention the age demographics of the younger age group [29,31]. Because of the inaccuracy of FITs and the considerably lower incidence of CRC in the younger group, identifying a novel modality may be necessary for the prevention of EOCRC in patients aged younger than 40 years. Second, the heterogeneity in this study was high for a meta-analysis. However, most pooled estimates remained robust after sensitivity analysis. Hence we considered a meta-analysis using this study to be convincing. Third, low-risk colorectal adenomas and sessile serrated lesions were not comprehensively reported on in many included studies, and the impact of polyp numbers and sidedness have not been explored by included studies, preventing further meticulous analysis. FIT is associated with lower sensitivity for right-sided lesions for average-risk individuals in previous studies [42,43]. The low sensitivity in FITs has also been reported for sessile serrated lesions [44] which represent up to one-fifth of CRCs [45]. However, the role of sessile serrated lesions in EOCRCs remains unclear. Last but not least, we could not suggest an optimal FIT cutoff value in the younger group because of the scarcity of data. Despite the outcomes regarding ACRN and CRC that may be adequate for judgement of the performance of FIT, future studies are necessary to clarity the above questions.

This systematic review and meta-analysis indicated that FITs may be a useful tool for the detection of ACRN in individuals aged 40–49 years and should be considered for use as a first-line screening tool from 45 years. Studies addressing cost-effectiveness and special risk factors are warranted to determine the optimal EOCRC screening tool choice.

## 5. Summary and Conclusions

The detection rate of ACRN and CRC based on FITs in individuals aged 40–49 years is acceptable, and the yield of ACRN might be similar between individuals aged 45–49 and 50–59 years. Further comparative and cost-effective analysis of colonoscopies is mandatory to guide its use in EOCRC screening.

## Figures and Tables

**Figure 1 cancers-15-03006-f001:**
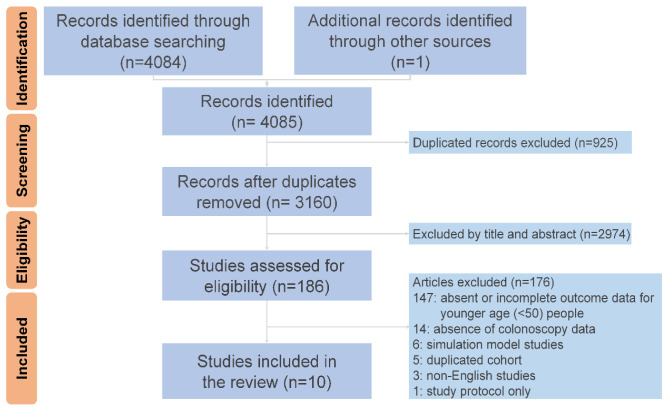
Preferred Reporting Items for Systematic Reviews and Meta-Analyses (PRISMA) flowchart.

**Figure 2 cancers-15-03006-f002:**
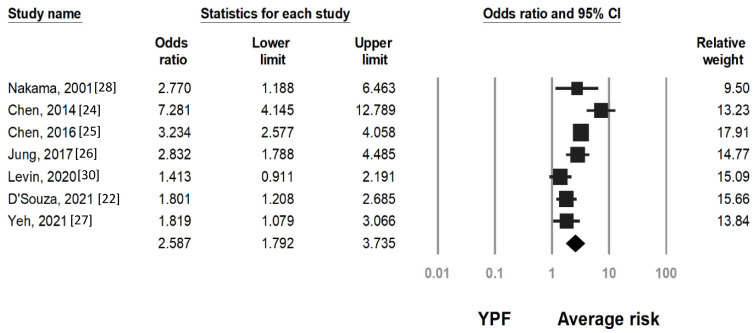
Pooled odds ratio (OR) of advanced colorectal neoplasm (ACRN) in younger individuals with positive fecal immunochemical test results (YPF) versus that of average-risk individuals. Heterogeneity: I^2^ = 79.2%, τ^2^ = 0.493183, *p* < 0.001.

**Figure 3 cancers-15-03006-f003:**
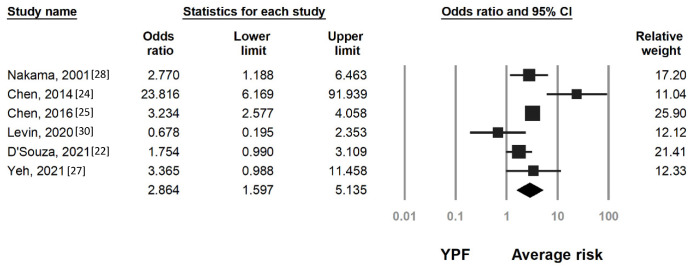
Pooled OR of colorectal cancer (CRC) of younger individuals with positive fecal immunochemical test results (YPF) versus that of average-risk individuals. Heterogeneity: I^2^ = 72.8%, τ^2^ = 0.329, *p* = 0.002.

**Figure 4 cancers-15-03006-f004:**
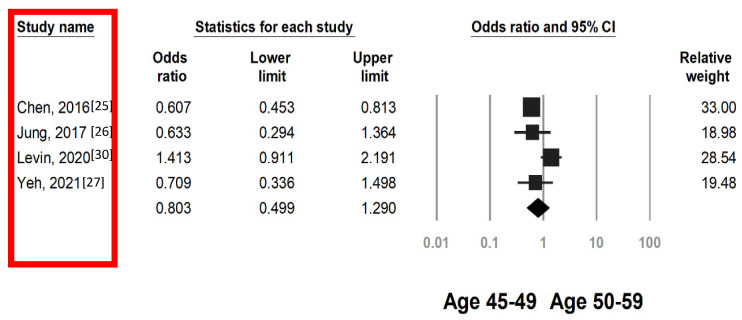
Pooled OR of ACRN of individuals with positive fecal immunochemical test results aged 45–49 versus 50–59 years. Heterogeneity: I^2^ = 70.5%, τ^2^ = 0.155, *p* = 0.017.

**Figure 5 cancers-15-03006-f005:**
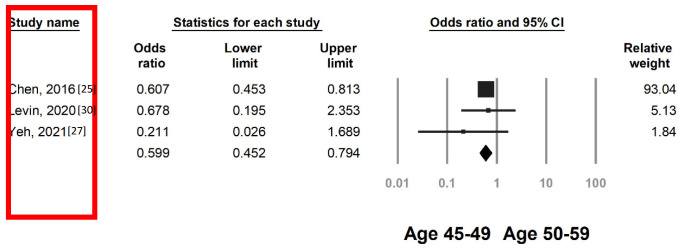
Pooled OR of CRC of individuals with positive fecal immunochemical test results aged 45–49 versus 50–59 years. Heterogeneity: I^2^ = 0, τ^2^ = 0, *p* = 0.602.

**Table 1 cancers-15-03006-t001:** Baseline characteristics of the included studies.

Study	Region	Patients by Age, N		Age Started to Screen	Gender (Male, %)	Product Name of FIT	Cutoff Value of FIT (µg Hb/g Feces)
		40–49	≥50				
Nakama, 2001 [28]	Japan	2382	5018	40	NA	OC-Hemodia	NA
Morikawa, 2007 [23]	Japan	12,696	9109	40	71.9	Magstream 1000 Hem SP	NA
Chen YY, 2014 [24]	Taiwan	2214	3882	40	56.0	OC-Light	10
Symonds, 2015 [29]	Australia	1849 ^†^	19,839	19	45.2	OC-Sensor	20
Chen CH, 2016 [25]	Taiwan	92,062	141,982	20	47.6	OC-Sensor	20
Jung, 2017 [26]	Australia	8819	2233	30	74.7	OC-Sensor Diana	20
Levin, 2020 [30]	US	3390	13,442	45	52.2	OC-Sensor Diana	20
D’Souza, 2021 [22]	England	1103 ^†^	8719	17	45.0	HM-Jack analyser	10
Pin-Vieito, 2021 [31]	Spanish	8866 ^†^	29,809	18	46.0	OC-Sensor^TM^	20
Yeh, 2021 [27]	Taiwan	1857	3150	20	57.6	OC-Sensor	20

FIT: fecal immunochemical test; ^†^: younger age group people aged <40 were also included.

**Table 2 cancers-15-03006-t002:** The characteristics of colorectal neoplasms among included studies.

Study ^§^	FIT Positive Patients by Age, N		Total Colono- scopy Exams by Age, N		Total CRN by Age, N		CRN of FIT Positive Patients by Age, N (%)		ACRN of FIT Positive Patients, by Age, N (%)		CRC of FIT Positive Patients, by Age, N (%)	
	40–49	≥50	40–49	≥50	40–49	≥50	40–49	≥50	40–49	≥50	40–49	≥50
Nakama, 2001 [28]	138	315	2382	5018	8	81	6 (4.3)	56 (17.8)	6 (4.3)	56 (17.8)	6 (4.3)	56 (17.8)
Morikawa, 2007 [23]	1231 (total cases)		12,696	9109	See footnotes below ^a^							
Chen YY, 2014 [24]	64	165	2214	3882	380	1096	25 (39.1)	78 (47.3)	18 (28.1)	38 (23.0)	3 (4.7)	6 (3.6)
Symonds, 2015 [29]	73 ^†^	1104	67 ^†^	1090	NA	NA	15 ^†^ (20.5)	512 (46.4)	NA	NA	NA	NA
Chen CH, ^‡§^ 2016 [25]	3728	3835	NA	NA	272	554	89 ^‡^ (2.3)	213 ^‡^ (5.5)	89 ^‡^ (2.3)	213 ^‡^ (5.5)	89 ^‡^ (2.3)	213 ^‡^ (5.5)
Jung, 2017 [26]	258	94	8819	2233	1634	643	NA	NA	26 (10.0)	20 (21.2)	NA	NA
Levin, 2020 [30]	136	575	116	451	NA	NA	67 (49.3)	298 (51.8)	39 (28.7)	119 (20.7)	3 (2.2)	17 (3.0)
D’Souza, 2021 [22]	212 ^†^	1674	1103	8719	189 ^†^	2882	NA	NA	29 ^†^ (13.7)	461 (27.5)	13 ^†^ (6.1)	286 (17.1)
Pin-Vieito, ^§^ 2021 [31]	754 ^†^	5900	NA	NA	See footnotes below ^b^							
Yeh, 2021 [27]	109	207	1857	3150	389	897	37 (33.9)	101 (48.7)	18 (16.5)	64 (30.9)	3 (2.7)	17 (8.2)

FIT: fecal immunochemical test, CRN: colorectal neoplasm, ACRN: advanced colorectal neoplasm, CRC: colorectal cancer, NA: not available, ^†^: younger age group people aged < 40 were also included, ^‡^: CRC cases were only extractable in 40–49 and 50–59 age groups. ^§^: only included colorectal cancer cases, ^a^: FIT has a sensitivity significantly higher than the false-positive rate to adenomas ≤ 9 mm (7.0% vs. 4.5%, *p* < 0.001), especially for men, ^b^: FIT sensitivity for CRC was 90.5% at a 10 mg Hb/g faeces threshold, and 87.4% at a 20 mg Hb/g feces threshold (additional CRC miss rate < 1/1000). The negative predictive value was more than 99% with any threshold.

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
