# Peer review of "Performance of the Fecal Immunochemical Test in Detecting Advanced Colorectal Neoplasms and Colorectal Cancers in People Aged 40–49 Years: A Systematic Review and Meta-Analysis"

_cancers, 2023, doi:10.3390/cancers15113006_

Round 1

Reviewer 1 Report

Manuscript:  Performance of the fecal immunochemical test in detecting advanced colorectal neoplasms and colorectal cancers in people aged 40–49 years: A systematic review and meta-analysis

General Comments
This systematic review and meta-analysis summarizes the evidence on the detection of advanced colorectal neoplasm and colorectal cancer among younger individuals by fecal immunochemical tests (FITs). This is a timely and interesting manuscript that may be relevant in addressing the increasing rate of colorectal cancer among younger adults. A few specific suggestions to improve the manuscript are included below.

Specific Comments
1)  Results, Table 1:  There is a wide range for the “age started to screen” among the included studies.  Although the authors mention that the study by D’Souza had a median age of 44 years, it would be helpful to also understand the characteristics of other studies that included participants much younger than age 40, such as the studies by Pin-Vieito and Symonds.

2) Results:  It seems like there is a mix-up of figures.  Supplementary Figures are referenced in the main text, but Schemes seem to be duplicates of the supplementary material.  Supplementary Figure 4 is also duplicated in the main text.

3) Results, line 169:  The OR 0.38, 95% CI 0.28-0.53 is not reflected in Supplementary Figure 2.

4) Results, lines 237 and 240:  Again, the listed results in the main text are not reflected in Supplementary Figures 4 and 5.

5) Discussion, lines 295-297:  Can the authors expand on the listed strengths?  It is unclear how conducting a systematic review and meta-analysis, or analyzing the age and participant characteristics, are strengths.

Quality of English language is fine, minor editing is needed.

Author Response

Please see the attachment below.

Reviewer 2 Report

Ø  “However, the efficacy of FITs in younger individuals is unclear.”

What is the efficacy of this test in individuals older than 50 in comparison to colonoscopy?

Ø  As you allude to in your introduction, EOCRC has a sidedness factor. Colonoscopies have been shown to be less effective in right than left colon. Do you see differences in your FIT analysis through a stratified approach? I.e. was FIT better at detecting EOCRC/ACRN that were identified in the left than the right colon?

Ø  Outcomes assessed include large adenoma size (line 96), was the total number of polyps detected at colonoscopy also considered?

Ø  Line 108: Why was a P value of < 0.1 considered for testing statistical heterogeneity?

Ø  Table 2 needs to be reformatted. Currently, there is text appearing at Morikawa and Pin-Vietio. Please removed sentences and include them in footnote or main text, if necessary.

Ø  Lines 231-242: this sensitivity analysis is helpful. However, it would be better placed as a sentence after each test, rather than as an independent section. Something along the lines of:

“Sensitivity analysis revealed…”

This stops the reader from having to go back to determine which of the results presented earlier should now be reconsidered.

Ø  Please check figure presentations. On line 244-245, the pdf generated by the journal has a figure legend for Supplementary Figure 4. Supplementary Figure Legends should be included at the end of the document.

Ø  Line 247: please change “Scheme” to “Figure”.

Ø  Line 250: please change “Scheme” to “Table”

Ø  African Americans present with the second highest burden for EOCRC in the US of all racial/ethnic groups. They also present with the highest overall burden for CRC. What was the demographic breakdown of the studies considered here and how was race factored into findings?

Author Response

Please see the attachment below.

Round 2

Reviewer 2 Report

Thank you for addressing my comments. I have nothing more to add.